# Unveiling the Uncertainty in Embodied and Operational Carbon of Large AI Models through a Probabilistic Carbon Accounting Model

**Xiaoyang Zhang[1], Fang He[1], Yang Deng[1], and Dan Wang[2]**

[1]Department of Computing, Hong Kong Polytechnic University
[2]Division of Environment and Sustainability, Hong Kong University of Science and Technology
`{xiaoyang.zhang,fangf.he,yang2.deng}@connect.polyu.hk,`
`wangdan@ust.hk`

## Abstract

The rapid growth of large AI models has raised significant environmental concerns due to their substantial carbon footprint. Existing carbon accounting methods for AI models are fundamentally deterministic and fail to account for inherent uncertainties in embodied and operational carbon emissions. Our work aims to investigate the effect of these uncertainties on embodied and operational carbon footprint estimates for large AI models. We propose a Probabilistic Carbon Accounting Model (PCAM), which quantifies uncertainties in the carbon accounting of large AI models. We develop parameter models to quantify key components (processors, memory, storage) in the carbon footprint of AI models. To characterize the distribution of the parameters, we develop a carbon dataset by aggregating related data from various sources. Then, we generate the probabilistic distribution of the parameters from the collected dataset. We compare the performance of PCAM with LLMCarbon, the state-of-the-art carbon accounting method for large AI models. PCAM achieves $\leq 7.44\%$ error compared to LLMCarbon's $\leq 108.51\%$.

## 1 Introduction

Large AI models have demonstrated high efficacy across various tasks. However, as model parameters and training datasets expand, the computational demands for training and deploying large-scale models grow significantly, leading to notable carbon emissions. For example, recent studies reveal that training advanced language models like Google's T5 generates 40% more carbon emissions than a transcontinental flight between San Francisco and New York [1]. The AI community is increasingly focused on developing methodologies to tackle social issues [2, 3, 4], particularly in promoting decarbonization and achieving carbon neutrality by 2050 [5, 6, 7, 8]. Carbon accounting, i.e., to estimate the carbon footprint of a product, is crucial for understanding the environmental impact of various operations and for developing strategies to reduce carbon footprints. By quantifying emissions, organizations can set reduction targets, comply with regulations, and demonstrate their commitment to sustainability. The carbon footprint of large AI models encompasses two components: operational carbon (i.e., emissions associated with electricity consumed) and embodied carbon (i.e., emissions associated with AI hardware manufacturing).

While existing studies have initiated carbon accounting for AI models [5, 1], current methodologies exhibit two fundamental limitations. First, they predominantly focus on operational carbon, while neglecting or applying a simple model to account for embodied carbon. They utilize coarse-grained average embodied carbon values for one hardware class (e.g., 14nm CPU) directly from Life Cycle Assessment (LCA) reports, potentially introducing substantial inaccuracies. Second, all existing

carbon models for AI models are single-deterministic and fail to capture the inherent uncertainty in the carbon footprint of AI models. The uncertainty in carbon accounting for AI models comes from (1) geotemporal differences in manufacturing, whereby identical device classes fabricated in different regions and periods embody distinct emissions due to different electricity carbon intensity during production; (2) efficiency evolution over time, as yield improvements, and fabrication energy efficiency gains; and (3) operating context variability, since the operational carbon of AI models can vary significantly depending on the location and time of the operation due to spatiotemporal variations in the carbon intensity of electricity.

In this paper, we propose a **P**robabilistic **C**arbon **A**ccounting **M**odel (PCAM), which can accurately estimate the carbon footprint of large AI models by capturing the uncertainty in the carbon modeling. Specifically, we develop parameter models for each hardware component (e.g., processors, memory, storage) in the carbon footprint of large AI models. To construct the distribution of the parameters, we make an effort to develop a hardware and electricity dataset by aggregating data from diverse sources, including Environmental, Social, and Governance (ESG) reports from hardware manufacturers, power grid operator statistics, industry reports, and peer-reviewed research publications. Then, we implement a simple yet effective dual-stage distribution modeling (i.e., converting the related collected data into frequency histograms followed by Kernel Density Estimation (KDE)), to generate the continuous probability density functions of the parameters.

We compare the performance of PCAM to LLMCarbon, the state-of-the-art carbon modeling method for large AI models. The evaluation is conducted based on four representative large AI models (XLM, T5, GPT-3, and Switch) [5, 9]. We compare the performance at key distribution percentiles (5th, 10th, 50th, 90th, and 95th) of PCAM's probability density function outputs against the ground truth based on the collected carbon dataset. Our PCAM outperforms LLMCarbon in both embodied and operational carbon on all large AI models. The results of PCAM show differences with the ground truth of only $\leq 2.67\%$ for embodied carbon and $\leq 7.44\%$ for operation carbon, demonstrating significantly higher accuracy compared to LLMCarbon ($\leq 23.02\%$ and $\leq 108.51\%$, respectively).

Our contributions can be summarised as follows:

- We propose a new Probabilistic Carbon Accounting Model (PCAM) for large AI models that can generate distribution-based outputs of carbon footprint instead of singular values, allowing AI developers to make risk-aware and sustainable decisions.

- We characterize the uncertainty in the carbon footprint of large AI models across key hardware components by generating the distributions of AI hardware-related parameters through a simple yet effective dual-stage distribution modeling from the collected dataset based on KDE.

- We make an effort to develop a carbon dataset [1] containing AI hardware-related parameters (e.g., yield, etc.) across different technology nodes from various technology reports, ESG reports, LCA reports, and carbon intensity of electricity data across regions from power grid operators.

## 2 Background

### 2.1 Large AI models carbon footprint

The carbon footprint of large AI models includes two components, i.e., operational carbon and embodied carbon. The operational carbon of large AI models refers to the emissions stemming from the electricity consumption in the model training or inference. The embodied carbon arises from the manufacturing processes of the hardware used to execute the AI models. The development of large AI models requires not only substantial electricity but also extensive computing hardware resources. For example, training and deploying large AI models require high-performance GPUs (e.g., NVIDIA A100 [10]) and ML accelerators (e.g., Google TPU [11]). These devices are typically manufactured with large processor chips using advanced technologies (e.g., 5nm), which results in significant carbon emissions. This refers to the embodied carbon of large AI models, which is especially notable in the models operated using the latest high-performance devices. As the proportion of green energy in the power grid increases and more and more data centers adopt carbon-free energy, the running

---

[1]Available at https://github.com/stuabc/PCAM

Table 1: Carbon emission factors (g/kWh) for energy sources [14]

| Emission factors | Oil | Coal | Natural gas | Nuclear | Wind | Solar | Hydro | Geothermal | Biomass | Other |
|---|---|---|---|---|---|---|---|---|---|---|
| Life-cycle emissions | 650 | 820 | 490 | 12 | 11 | 45 | 24 | 38 | 230 | 700 |

carbon of large AI models will be greatly reduced, resulting in embodied carbon accounting for a non-negligible part of the large AI models' carbon footprint [12, 13].

## 2.2 Carbon emission factor and carbon intensity

The *carbon emission factor* (in g/kWh) is defined as the quantity of carbon emission per unit of electricity generated by a specific energy source. The carbon emission factors of brown sources (e.g., coal, gas, etc.) are much higher than those of green sources (e.g., wind, solar, etc.), as Table 1 shows.

The *carbon intensity* of electricity is the carbon emission rate (in $g/kWh$) when the electricity is generated, i.e., the total amount of carbon emitted ($Gram$) as against the electricity generation ($Kilowatt - Hour$). It is the weighted average of carbon emissions by each energy source due to the electricity generated by them. Mathematically, the carbon intensity of electricity generated at any time is as follows:

$$Carbon\ Intensity = \frac{\sum ef^k \times E^k}{\sum E^k} \qquad (1)$$

where $ef^k$ is the carbon emission factor and $E^k$ is the electricity generated by energy source $k$.

## 3 Methodology

### 3.1 Limitation of existing modeling techniques

The existing carbon modeling techniques for AI models can be summarized as equation 2, where $C_{model}$ denotes the total carbon footprint of the AI models, including embodied carbon ($EC_{model}$) and operational carbon ($OC_{model}$), $t$ is the time the model occupies the hardware system, $T$ is the lifetime of the hardware system, $E$ is the electricity consumed by the hardware system, and $CI$ is the carbon intensity of electricity.

$$C_{model} = EC_{model} + OC_{model} = t/T \cdot EC_{system} + E \cdot CI \qquad (2)$$

These existing modeling techniques for AI models' carbon footprint are fundamentally deterministic and estimate the carbon emission model as a single deterministic value. These methods typically rely on lifecycle assessment (LCA) reports that generalize embodied carbon estimates ($EC_{system}$) across entire product categories (e.g., 28nm CPUs, LPDDR4 DRAM). However, this paradigm fails to account for critical spatial, temporal, and manufacturing variability that substantially impacts actual carbon emissions. Specifically, the existing carbon models for AI models fail to capture: (1) geotemporal differences in manufacturing, whereby the device instances in identical device classes are fabricated in different regions and periods, e.g., in the winter in Ireland (Intel) or in the summer in Taiwan (TSMC), embody distinct emissions due to different electricity carbon intensity during production; It is known that carbon intensity depends on the electricity generation process, which has spatial and temporal dynamics; (2) efficiency evolution over time, arising from continuous advancements in process yield and the energy efficiency of fabrication methods; and (3) operating context variability, since the operational carbon of AI models can vary significantly depending on the when and where the AI models depoly because of spatiotemporal variations in the carbon intensity of electricity. These characteristics introduce uncertainty into carbon estimates for AI models; however, current carbon accounting methods for AI models fail to account for these uncertainties.

### 3.2 Embodied carbon modeling

We extend the existing carbon modeling methodology of AI models by giving parameter models for each hardware component and characterizing the uncertainty. Specifically, we analyze the parameter

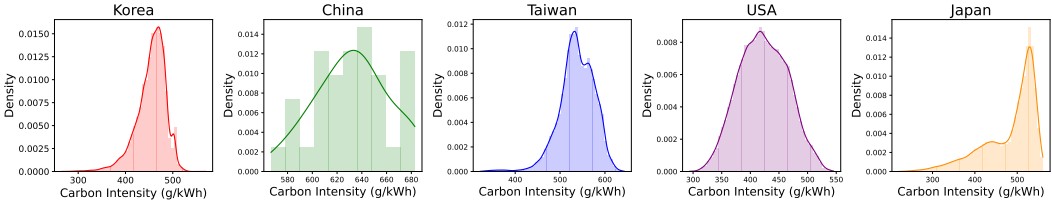

Figure 1: Histograms of hourly carbon intensity data (monthly data in China) from five major IC production regions between 2021-2023 with their individual kernel density estimates (KDEs).

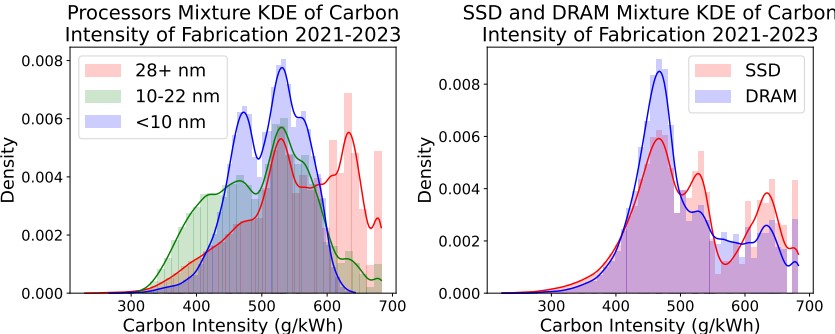

Figure 2: The distribution of carbon Intensity of processors, memory (DRAM), and storage (DRAM) fabrication 2021-2023.

value evolution in the fabrication process by collecting data from various sources (e.g., ESG reports, technology reports, power grid operators, etc.). We leverage KDE to generate the distributions of the parameters based on the data we collect. KDE is a standard non-parametric estimation technique used to estimate the probability density function of a random variable by applying a Gaussian kernel weighting technique to data [15]. It is used when the behavior of a sparse data sample is extrapolated to resample data to a larger sample size.

### 3.2.1 Embodied carbon of processors

We model the embodied carbon of large AI models caused by processors as follows: (1) the operation time of large AI models on processors, denoted as $t_p$, and the processor's lifetime, denoted as $T_p$. (2) the carbon emissions from electricity used during the fabrication process, calculated by multiplying the Electricity consumption Per unit of die Size ($EPS$) by the carbon intensity of electricity during manufacturing $CI_m$; (3) the emissions associated with the raw materials, denoted as $MPS$ (Material Per Size); and (4) the carbon released by gas (e.g., fluorinated compounds) in the manufacturing process denoted as $GPS$ (Gas Per Size); (5) the die size of the processor, denoted as $DizeSize$; (6) fab yield denoted as $Y$. Then, we can model the embodied carbon of processors for large AI models as follows:

$$EC_{model}^{p} = \frac{t_p \cdot DieSize}{T_p \cdot \tilde{Y}} \cdot (\tilde{CI_m} \cdot \tilde{EPS} + GPS + MPS) \tag{3}$$

Here, we denote $\tilde{Y}$ as the probabilistic model of yield $Y$. Similarly, we have $\tilde{CI_m}$ and $\tilde{EPS}$. We will give the detailed uncertainty characterization of the three parameters in the following.

*Carbon intensity distribution in the manufacturing process.*

$CI_m$ denotes the carbon intensity of electricity consumed during semiconductor manufacturing, a critical parameter determined by the energy source mix (e.g., solar, wind, coal, etc.) employed in electricity generation. This metric exhibits spatiotemporal variability due to fluctuations in both manufacturing schedules and geographic locations. The inherent uncertainty in carbon intensity primarily comes from temporal variations throughout annual cycles, driven by seasonal patterns in renewable energy generation and fluctuating electricity demand. When the manufacturing location is specified, regional fabrication facilities demonstrate distinct carbon intensity distribution derived from historical carbon intensity data from 2021 to 2023, as illustrated in Figure 1.

Table 2: Global wafer fabrication capacity by technology by regions in 2020.

| Regions | US | China | Taiwan | Korea | Japan | Others |
|---|---|---|---|---|---|---|
| <10nm Logic | 0% | 0% | 69% | 31% | 0% | 0% |
| 10-22nm Logic | 28% | 6% | 40% | 9% | 5% | 12% |
| 28nm+ Logic | 8% | 33% | 30% | 5% | 10% | 14% |
| Memory(DRAM) | 3% | 18% | 20% | 52% | 7% | 0% |
| Storage(NAND) | 3% | 26% | 4% | 30% | 30% | 7% |

Memory can usually be manufactured by multiple manufacturers (e.g., SK Hynix, Samsung, Micron, etc.), unlike processor manufacturers, which have a monopoly (e.g., most GPUs are manufactured by TSMC). For the scenario involving uncertain manufacturing locations, the region selection can be modeled as a discrete random variable. The probability distribution across geographical regions is weighted according to their respective contributions to global IC production capacity for specific process nodes [16]. As detailed in Table 2 from industry reports [17], these capacity allocations serve as probabilistic weights for regional selection. The composite probability distribution for each process node is constructed through a mixture modeling approach, incorporating the following components: (1) regional KDE: non-parametric distributions developed using historical carbon intensity data (2021-2023) for each major production region; (2) capacity-weighted sampling: a Monte Carlo sampling strategy where regional selection probabilities correspond to normalized production capacities; (3) composite distribution formation: aggregation of weighted regional samples through kernel density smoothing, as visualized in Figure 2. This enables probabilistic modeling of carbon intensity while accounting for both geographical production distributions and temporal energy mix variations.

*Yield Distribution.* Semiconductor manufacturing yield is defined as the ratio of defect-free semiconductor dies on a wafer to the total number of dies on that wafer. The parameter exhibits inherent uncertainty that stems primarily from temporal variations in defect density observed across fabrication facilities. To systematically investigate this phenomenon, we employ TSMC's historical defect density data across four distinct process nodes [18], coupled with the application of the Poisson yield model [19] for temporal yield computation. Following empirical data collection, we generate yield histograms and subsequently apply KDE [20] to derive comprehensive probability density functions characterizing yield distributions per unit die size, as visually presented in Figure 3.

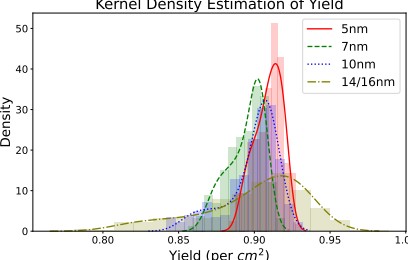

Figure 3: The Kernel Density Estimation of Yield $\tilde{Y}$.

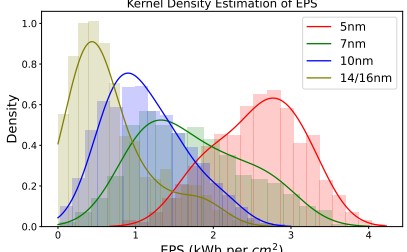

Figure 4: The kernel density estimation of $\tilde{EPS}$ across different technology nodes.

*EPS Distribution.* The inherent uncertainty in $EPS$ is from temporal fluctuations in process energy efficiencies across different fabrication stages. We build the EPS distributions based on the EPS values from the STEC model [13], the ACT model [12], the imec model [21], and the annual energy efficiency improvements data from across the TSMC ESG reports [22] across various technology process nodes (e.g., 5nm, 7nm). To construct the distributions, the analytical procedure involved three key transformations: First, energy efficiency metrics for each process node are temporally normalized relative to their baseline year values. Second, raw $EPS$ values are adjusted through division by these normalized efficiency coefficients to account for technological advancements. Finally, we implement a dual-stage distribution modeling approach, i.e., converting the processed data into frequency histograms followed by KDE to derive continuous probability density functions of $\tilde{EPS}$, as visualized in Figure 4.

### 3.2.2 Embodied carbon of memory

We model the embodied carbon of large AI models caused by memory as follows: (1) the operation time of large AI models on memory $t_m$ and the memory's lifetime $T_m$; (2) the carbon released by the electricity consumed during the process of manufacturing the memory. This is calculated by multiplying the electricity consumption per unit of size ($EPS$) by the carbon intensity of a power grid ($CI_m$) then divided by bit density ($BD$, in $GB/mm^2$); (3) the carbon released independent of electricity, such as raw materials, distribution, and packaging, denoted as $\alpha_m$; and (4) the capacity of memory $C_m$. Then, we can model the embodied carbon of memory for large AI models as follows:

$$EC^m_{model} = t_m/T_m \cdot C_m \cdot (\tilde{CI}_m \cdot \tilde{EPS}/BD + \alpha_m) \tag{4}$$

Here, the distribution of $\tilde{CI}_m$ and $\tilde{EPS}$ for memory is similar to that in processors. We develop the distribution based on the data we collect from Hynix [23] as Figure 2 shows.

### 3.2.3 Embodied carbon of storage

We model the embodied carbon of large AI models caused by storage as follows: (1) the operation time of large AI models on storage $t_s$ and the storage's lifetime $T_s$; (2) the carbon emissions from the process of manufacturing the storage, which is determined by multiplying the electricity consumed per unit of Giga-Byte during the manufacturing process ($EPG$) by the carbon intensity $CI_m$; (3) the carbon released independent of electricity such as raw materials, denoted as $\alpha_S$. $\alpha_S$ can be found in the industry reports [24]; and (4) the capacity of storage, denoted as $C_s$. Then, we can model the embodied carbon of storage for large AI models and omit the distribution derivation as follows:

$$EC^s_{model} = t_s/T_s \cdot C_s \cdot (\tilde{CI}_m \cdot EPG + \alpha_S) \tag{5}$$

### 3.3 Operational carbon

The operational carbon footprint of large AI models ($OC_{model}$) refers to the carbon emissions caused by the electricity used during operation. It is determined by the electricity consumed by the model ($E_o$) and the carbon intensity of the electricity ($CI_o$) as follows:

$$OC_{model} = E_o \cdot \tilde{CI}_o \tag{6}$$

where $\tilde{CI}_o$ represents the distribution of the carbon intensity of that electricity. This uncertainty comes from the uncertainty of the time and location of model training or inference. $E_o$ can be recorded by hardware during the model operation, or it can be estimated by the following equations:

$$E_o = \sum_{i \in HardwareSet} (P_i \cdot eff_i \cdot n_i \cdot t_i) \cdot PUE \tag{7}$$

where $P_i$ represents the peak power of hardware $i$; $eff_i$ denotes the efficiency of hardware $i$ that can be estimated by the hardware efficiency model [1]; $n_i$ is the number of hardware $i$; and $t_i$ refers to the execution time for hardware $i$ that can be estimated according to the existing flop model [1]. These hardware units include CPUs, GPUs, memory, storage, and others.

## 4 Evaluation

### 4.1 Setup

To comprehensively evaluate PCAM for the carbon footprint of large AI models, we make an effort to develop a hardware and electricity dataset. This dataset includes hardware-related parameters (e.g., EPS, EPG, MPS, BD, etc.) and electricity-related parameters (e.g., CI), which is constructed by aggregating data from diverse sources, including ESG reports from hardware manufacturers, power grid operator statistics, industry reports, and peer-reviewed research publications

Table 3: The hardware information about XLM.

| Hardware | Number | Die Size/Unit | Technology Nodes |
|---|---|---|---|
| GPU | 512 | 8.15 cm2 | 12nm |
| CPU | 64 | 1.47 cm2 | 16nm |
| Storage(SSD) | 64 | 32TB | Seagate Nytro 3332 |
| Memory(DRAM) | 64 | 256GB | 10nm ddr4 |

For performance evaluation, we conduct a comparative analysis between PCAM and LLMCarbon, the current state-of-the-art carbon model for large AI models. The evaluation is performed using published training data from four representative large AI models (XLM, T5, GPT-3, and Switch) [5, 9]. We compare the performance at key distribution percentiles (5th, 10th, 50th, 90th, and 95th) by resampling PCAM's probability distribution outputs against the ground truth based on the collected carbon dataset. The evaluation focuses on the training processes of large AI models, as the accounting methodology is identical for both training and inference. Please note that the carbon footprint can not be measured by sensors due to its inherent characteristic. The "ground truth" in the following evaluation is the accounting result based on the carbon dataset we collected without KDE.

## 4.2 Embodied carbon evaluation

We conduct a comparative analysis of embodied carbon accounting between PACM and LLMcarbon, utilizing the published XLM training data [5]. Notably, to the best of our knowledge, the XLM dataset represents the only publicly available information detailing hardware infrastructure associated with embodied carbon for large AI models training. The hardware configuration, comprising 64 servers with detailed specifications presented in Table 3, is employed for XLM's training process. The training duration lasts 20.4 days, with all hardware components assumed to have a standard operational lifetime of 5 years [5].

The embodied carbon accounting performance of PCAM and LLMCarbon is compared across different hardware components and statistical percentiles, as detailed in Table 4. For total embodied carbon, PCAM demonstrates great alignment with ground truth values across all percentiles, maintaining a maximum deviation of only 2.67% at the 10th percentile and achieving remarkable precision at the median (0.48% deviation). In contrast, LLMCarbon shows substantially larger deviations ranging from 6.03% at the median to 23.02% at the 5th percentile, demonstrating that the single deterministic model fails to capture uncertainty in embodied carbon accounting.

Table 4: The comparison between LLMCarbon and PCAM on embodied carbon accounting.

| Models | Total Embodied Carbon (kg) at each Percentile | | | | |
|---|---|---|---|---|---|
| | 5th | 10th | Median | 90th | 95th |
| Ground Truth | 229.17 | 236.27 | 279.75 | 331.32 | 352.15 |
| PCAM | 235.2 | 242.74 | 281.09 | 339.95 | 359.93 |
| PCAM △ | 2.56% | 2.67% | 0.48% | 2.54% | 2.16% |
| LLMCarbon | | | 297.71 | | |
| LLMCarbon△ | 23.02% | 20.64% | 6.03% | 11.29% | 18.29% |
| | CPU Embodied Carbon (kg) at each Percentile | | | | |
| Ground Truth | 1.14 | 1.18 | 1.49 | 2.04 | 2.18 |
| PCAM | 1.16 | 1.19 | 1.48 | 2.01 | 2.12 |
| PCAM△ | 1.72% | 0.81% | 0.52% | 1.45% | 2.61% |
| LLMCarbon | | | 1.55 | | |
| LLMCarbon△ | 26.50% | 23.89% | 4.08% | 31.47% | 40.25% |
| | GPU Embodied Carbon (kg) at each Percentile | | | | |
| Ground Truth | 98.69 | 101.36 | 133.76 | 189.11 | 211.05 |
| PCAM | 96.97 | 99.42 | 132.29 | 187.62 | 209.39 |
| PCAM△ | 1.77% | 1.95% | 1.11% | 0.79% | 0.79% |
| LLMCarbon | | | 141.68 | | |
| LLMCarbon△ | 66.85% | 28.46% | 5.59% | 33.48% | 48.96% |
| | DRAM Embodied Carbon (kg) at each Percentile | | | | |
| Ground Truth | 7.97 | 8.15 | 10.39 | 14.47 | 15.3 |
| PCAM | 7.83 | 8.07 | 10.26 | 14.34 | 15.21 |
| PCAM△ | 1.79% | 0.99% | 1.27% | 0.91% | 0.59% |
| LLMCarbon | | | 10.85 | | |
| LLMCarbon△ | 26.54% | 24.88% | 4.24% | 33.36% | 41.01% |
| | SSD Embodied Carbon (kg) at each Percentile | | | | |
| Ground Truth | 105.99 | 110.23 | 123.62 | 145.17 | 147.74 |
| PCAM | 108.25 | 111.47 | 122.76 | 147.05 | 149.65 |
| PCAM△ | 2.09% | 1.11% | 0.70% | 1.28% | 1.28% |
| LLMCarbon | | | 125.64 | | |
| LLMCarbon△ | 15.64% | 12.27% | 1.61% | 15.54% | 17.59% |

**Component-wise analysis** shows that PCAM outperforms LLMCarbon at each component. For CPU carbon accounting, PCAM errors are below 2.62% versus LLMCarbon's around 4% - 40% deviations. GPU calculations show PCAM's robust performance with less than 2% deviations across all percentiles, contrasted by LLMCarbon's substantial overestimations reaching around 67% at the 5th percentile. The DRAM and SSD comparison follows similar trends, with PCAM errors remaining

Table 5: The comparison between LLMCarbon and PCAM on different large AI models for operational carbon accounting.

| AI models | | Carbon Models | Operational Carbon in the Spatial Dimension (ton) | | | | | Operational Carbon in the Temporal Dimension (ton) | | | | |
|---|---|---|---|---|---|---|---|---|---|---|---|---|
| | | | 5th per. | 10th per. | Median | 90th per. | 95th per. | 5th per. | 10th per. | Median | 90th per. | 95th per. |
| XLM | Training Day: 20.4; PUE: 1.1; Ave. Power: 342 kw; Num. of device: 512 | Ground Truth | 2.87 | 4.93 | 28.19 | 56.77 | 62.74 | 33.41 | 34.67 | 39.78 | 45.07 | 46.41 |
| | | LLMCarbon | | | 30.09 | | | | | 38.86 | | |
| | | LLMCarbon Δ | 90.46% | 83.62% | 6.31% | 88.67% | 108.51% | 16.18% | 13.02% | 2.37% | 13.07% | 16.43% |
| | | PCAM | 2.79 | 5.13 | 29.08 | 57.13 | 63.71 | 33.59 | 34.19 | 39.07 | 45.51 | 46.03 |
| | | PCAM Δ | 2.87% | 3.90% | 3.06% | 0.63% | 1.52% | 0.54% | 1.40% | 1.82% | 0.97% | 0.83% |
| T5 | Training Day: 20; PUE: 1.12; Ave. Power: 310 kw; Num. of device: 512 | Ground Truth | 2.6 | 4.46 | 25.51 | 51.36 | 56.77 | 30.23 | 31.37 | 35.99 | 40.78 | 41.99 |
| | | LLMCarbon | | | 27.23 | | | | | 37.07 | | |
| | | LLMCarbon Δ | 90.45% | 83.62% | 6.32% | 88.62% | 108.48% | 16.19% | 13.03% | 2.91% | 13.06% | 16.41% |
| | | PCAM | 2.42 | 4.74 | 25.68 | 51.19 | 57.13 | 30.91 | 31.98 | 36.85 | 40.13 | 42.83 |
| | | PCAM Δ | 7.44% | 5.91% | 0.66% | 0.33% | 0.63% | 2.20% | 1.91% | 2.33% | 1.62% | 1.96% |
| GPT3 | Tarinin Day: 14.8; PUE:1.1; Ave. Power: 330 kw; Num. of device:10K | Ground Truth | 39.36 | 67.53 | 385.49 | 776.22 | 857.85 | 456.83 | 474.09 | 543.94 | 616.31 | 634.55 |
| | | LLMCarbon | | | 411.49 | | | | | 559.09 | | |
| | | LLMCarbon Δ | 90.43% | 83.59% | 6.32% | 88.64% | 108.47% | 16.19% | 13.03% | 2.71% | 13.07% | 16.41% |
| | | PCAM | 37.75 | 69.71 | 390.21 | 769.02 | 869.37 | 449.87 | 469.99 | 535.68 | 605.95 | 626.12 |
| | | PCAM Δ | 4.26% | 3.13% | 1.21% | 0.94% | 1.33% | 1.55% | 0.87% | 1.54% | 1.71% | 1.35% |
| Switch | Tarinin Day: 27; PUE:1.1; Ave. Power: 245 kw; Num. of device: 1K | Ground Truth | 5.33 | 9.14 | 52.21 | 105.13 | 116.19 | 61.87 | 64.21 | 73.67 | 83.47 | 85.94 |
| | | LLMCarbon | | | 55.73 | | | | | 74.82 | | |
| | | LLMCarbon Δ | 90.44% | 83.60% | 6.32% | 88.64% | 108.49% | 16.19% | 13.02% | 1.54% | 13.07% | 16.42% |
| | | PCAM | 5.01 | 9.49 | 52.23 | 105.04 | 118.09 | 61.18 | 64.88 | 73.08 | 83.95 | 85.08 |
| | | PCAM Δ | 6.39% | 3.69% | 0.04% | 0.09% | 1.61% | 1.13% | 1.03% | 0.81% | 0.57% | 1.01% |

below 1.8% in DRAM (2.1% in SSD) versus LLMCarbon's around 25% - 40% in DRAM (around 2% - 18% in SSD) deviations.

Our analysis reveals that GPUs exhibit a substantially wider absolute uncertainty range (98.69 - 211.05 $kg$, $\Delta$= 114.36 $kg$) in embodied carbon compared to the cumulative variance of other hardware components, establishing them as the dominant contributor to total embodied carbon variability. This disproportionate impact stems from two key manufacturing characteristics: First, GPU production employs advanced semiconductor fabrication processes that require significantly higher energy, making its embodied carbon particularly sensitive to variations in the carbon intensity of electricity. Second, the combination of larger die sizes and complex architectures results in lower yields with greater variation, amplifying the uncertainty in the embodied carbon of GPUs.

Overall, this significant variance underscores critical limitations in conventional deterministic carbon accounting models like LLMCarbon and highlights the necessity of probabilistic modeling that explicitly accounts for parameter uncertainties in carbon modeling. These findings challenge the validity of current sustainability certification paradigms that rely on deterministic embodied carbon accounting. We propose that comprehensive environmental impact assessments for large AI models should adopt distribution-aware evaluation methodologies and report confidence intervals alongside central estimates to enable more informed decision-making in sustainable large AI models development.

### 4.3 Operation carbon evaluation

Table 5 presents the operation carbon evaluation results of PCAM for the four large AI models, XLM, T5, GPT3, and Switch, based on their published training information [9]. The training information is listed in Table 5, where "avg. power (W)" conveys the average system power per computing device (e.g., GPU, TPU, DRAM, etc.); "Num. of device" is the total number of computing devices. We estimate the electricity consumption of large AI models based on the training data and subsequently calculate their operational carbon footprint.

To comprehensively evaluate model performance, we conducted comparative analyses across spatial and temporal dimensions. In the spatial analysis, we maintain a fixed training date while considering global geographical variations (across 90 regions). Conversely, the temporal analysis fixed the training location (the USA) while accounting for time-dependent factors (across 2021-2023 years). Overall, PCAM outperforms LLMCarbon (0.09% - 7.44% vs. 1.54%- 108.51% deviations) in the operational carbon accounting for each AI model. Besides, the absolute error of the LLMCarbon is much larger than that of PCAM, e.g., 372.13 $t$ vs. 3.6 $t$ for GPT3 under the 5th percentile in the spatial dimension.

Besides, spatial variations demonstrate particularly pronounced differences. For example, the deviations of LLMCarbon under the spatial dimension are from 90.43% (5th percentile) to 108.51% (95th percentile), while that is from 16.18% to 16.43% under the temporal dimension.

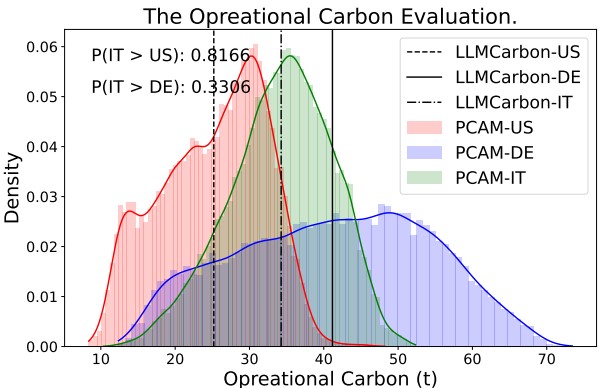

Figure 5: The comparison between LLMCarbon and PCAM on operational carbon accounting in the US, IT, and DE.

This contrast highlights that geographical factors introduce greater uncertainty in operational carbon accounting than temporal variations.

## 5 Case Study

PCAM delivers practical value by enabling risk-aware decision-making through carbon budget overrun probability assessments, ensuring realistic sustainability targets. It facilitates sensitivity analysis to pinpoint key uncertainty drivers, guiding operational and hardware prioritization while aiding manufacturers in green production optimization. For operational choices, PCAM helps developers schedule computing tasks based on temporal and regional grid carbon intensity. It also supports multi-source hardware comparisons by evaluating embodied carbon distributions to inform procurement. Furthermore, PCAM encourages industry-wide engagement by quantifying supply chain impacts and promoting transparency. Its probabilistic reporting enhances credibility for compliance while aligning with carbon-aware computing trends by providing confidence bounds for auditing and long-term planning.

Figure 5 provides a visual comparison of operational carbon estimates between PCAM and LLM-Carbon for XLM training in three regions: California (US), Germany (DE), and Italy (IT). While LLMCarbon suggests the US produces lower emissions than IT and IT has lower emissions than DE, PCAM reveals more nuanced probabilistic outcomes: there is an 81.66% probability of IT's operational carbon footprint exceeding that of the US and a 33.06% probability of surpassing DE's emissions. This probabilistic approach enables PCAM to provide multi-dimensional insights, empowering AI developers to make more informed sustainability decisions.

## 6 Related Work

The current methodologies for carbon footprint estimation of large AI models predominantly concentrate on operational carbon, which is calculated as the product of electricity consumed and carbon intensity. Most studies predominantly focus on tracking or estimating the electricity consumed. A subset of studies has developed software-integrated instrumentation to monitor real-time CPU/GPU power utilization during inference or training phases [25, 26, 27, 28], while alternative approaches derive energy estimates through the parameters of and hardware (e.g., thermal design power) [29, 30]. However, these methodologies systematically neglect the embodied carbon accounting associated with AI hardware infrastructure, which is non-negligible in the carbon footprint of large AI models, as power grids decarbonize and data centers increasingly adopt zero-carbon energy sources.

Regarding embodied carbon modeling, existing frameworks such as SustainableAI [5] estimated the embodied carbon of large AI models through manufacturer-reported emission factors for hardware components, which is an average value. LLMCarbon [1] adopts a deterministic parametric model to estimate the embodied carbon for processors and average values for storage and memory in ESG

reports. However, these approaches lead to oversimplified modeling constrained by two critical limitations: (1) fail to capture the effect of temporal and spatial specificity in hardware instance (e.g., 28nm CPU made in winter in China vs. 28nm CPU made in summer in the US); (2) fail to capture the effect of heterogeneous manufacturing configurations within device class (e.g., yield and fabrication efficiency improvement for certain processors). These characteristics lead to uncertainty in carbon estimates in large AI models, but current large AI models' carbon models fail to capture this uncertainty.

# 7    Limitations and Discussions

This study examines the carbon footprint of large AI models across four life-cycle stages: (1) Hardware manufacturing, referring to emissions generated during the production of core computing components; (2) Hardware transport, encompassing emissions from shipping hardware to end-users; (3) Operational use, comprising emissions resulting from software execution, primarily due to electricity consumption; and (4) End-of-life processing, involving emissions associated with hardware recycling. Among these, the operational use phase corresponds to the operational carbon of large AI models, while the remaining stages contribute to their embodied carbon. Hardware manufacturing and operational use are the dominant sources of emissions, for instance, over 97% in the case of a Dell R740 server [31]. In contrast, transport and end-of-life stages make negligible contributions and are thus excluded from the current scope. Within hardware manufacturing, we focus specifically on emissions from computer hardware production, such as those arising from material extraction, energy use, and chemical processing, while ancillary factors like infrastructure, cooling systems, and human labor are not considered in this study. We acknowledge that these omitted factors also play a significant role in the overall embodied carbon footprint. A full quantification of their associated uncertainties requires interdisciplinary collaboration and is reserved for future work.

PCAM only considers the uncertainty of some parameters due to the lack of relevant carbon data. Inaccuracy comes from using averaged/aggregated data in various aspects (e.g., raw material, supply chain, packaging, etc.). Intrinsically, carbon data is unique in that it cannot be directly measured by sensors (unlike electricity) and can only be calculated through accounting models, which leads to the fact that perfect carbon modeling is difficult. The increasing focus on carbon emissions considerations among hardware producers drives substantial enhancements in carbon data quality. This trend leads to more accurate and reliable accounting models for the carbon footprint of large AI models.

# 8    Conclusion

In this paper, we propose PCAM, a Probabilistic Carbon Accounting Model for large AI Models, which systematically captures and quantifies the uncertainties inherent in both embodied and operational carbon emissions. Unlike existing deterministic approaches such as LLMCarbon, PCAM provides distribution-based carbon footprint estimates, enabling AI developers and policymakers to make risk-aware decisions. By moving beyond point estimates to probabilistic carbon accounting, PCAM supports more robust environmental impact assessments, facilitates compliance with emerging carbon regulations, and promotes transparency in AI sustainability reporting. Future work will expand PCAM to incorporate additional lifecycle stages and supply chain factors, fostering a more holistic and actionable framework for carbon-aware AI development. To facilitate the decarbonization of large AI models, we are open-sourcing the carefully curated carbon dataset compiled in this study, thereby providing the community with the necessary tools to quantify, understand, and mitigate the environmental impact of AI model development. We strongly call for collective action across the industry to enhance the transparency of carbon-related data, including the carbon intensity of power grids and hardware-related data. Establishing a culture of open and standardized emissions reporting is fundamental to driving systemic decarbonization. It enables the establishment of credible baselines, pinpoints optimization hotspots, and paves the way for accountable and verifiable progress across the entire AI ecosystem.

# Acknowledgements

Dan Wang's work is supported in part by RGC GRF 15200321, 15201322, 15230624, 15239925, ITC ITF-ITS/056/22MX, ITS/052/23MX, and PolyU 1-CDKK, G-SAC8, K-ZYAP.

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

Table 6: The comparison between PCAM and STEC on the embodied carbon accounting of XLM.

| Models | Total Embodied Carbon (kg) at each Percentile | | | | |
|---|---|---|---|---|---|
| | 5th | 10th | Median | 90th | 95th |
| Ground Truth | 229.17 | 236.27 | 279.75 | 331.32 | 352.15 |
| PCAM | 235.20 | 242.74 | 281.09 | 339.95 | 359.93 |
| PCAMΔ | 2.56% | 2.67% | 0.48% | 2.54% | 2.16% |
| STEC | 283.01 | 286.00 | 294.90 | 303.02 | 305.01 |
| STECΔ | 23.49% | 21.05% | 5.42% | 8.55% | 13.39% |
| LLMCarbon | | | 297.71 | | |
| LLMCarbonΔ | 23.02% | 20.64% | 6.03% | 11.29% | 18.29% |

## Appendix

We provide additional experiments to compare PCAM with STEC [13], which is an embodied carbon accounting model for computer systems. STEC incorporates spatiotemporal variability in the carbon intensity of electricity used during hardware manufacturing. However, it remains a fundamentally deterministic model and does not probabilistically account for other key sources of uncertainty, such as geotemporal manufacturing capacity distribution or efficiency evolution over time (e.g., yield and fabrication efficiency improvements). To enable a direct quantitative comparison with PCAM at the AI model level, we extended the STEC methodology by scaling its hardware-level carbon accounting using the foundational equation (Eq. 2 from the main text) and generating a distribution of embodied carbon based on the regional electricity carbon intensity frequency data. PCAM outperforms STEC consistently across all percentiles, with errors under 3% compared to STEC's under 24%, as Table 6 shows.

