# OpenReview forum: "Unveiling the Uncertainty in Embodied and Operational Carbon of Large AI Models through a Probabilistic Carbon Accounting Model"
_NeurIPS.cc/2025/Conference — NeurIPS 2025 poster_

### Official Review · Reviewer_rNJZ · 2025-06-25

**Clarity:** 3
**Significance:** 3
**Originality:** 2
**Rating:** 4
**Confidence:** 3

**Summary:**

This paper addresses the critical issue of carbon emissions associated with large AI models, encompassing both operational and embodied carbon footprints. While prior studies have largely relied on deterministic estimates, this work identifies and models three key sources of uncertainty that affect carbon accounting: (1) geotemporal variability in manufacturing, (2) temporal evolution of manufacturing practices, and (3) dynamic operational conditions. To capture these factors, the authors propose a Probabilistic Carbon Accounting Model (PCAM) that integrates a dual-stage distribution modeling approach using Kernel Density Estimation (KDE) on a curated dataset of carbon-relevant parameters from diverse sources. Experimental results demonstrate that PCAM significantly improves estimation accuracy, achieving ≤ 7.44% error compared to LLMCarbon’s ≤ 108.51%.

**Questions:**

1. Zhang et al. [12] present a spatial-temporal model for embodied carbon that also addresses uncertainty, making it a strong baseline for comparison. Could you clarify why this work was not included in your experimental evaluation? Additionally, is there a technical limitation or incompatibility that prevents a direct quantitative comparison between PCAM and [12]? Including such a comparison would significantly strengthen your claims regarding PCAM’s accuracy and novelty.

**Ethical Concerns:**

["NO or VERY MINOR ethics concerns only"]

**Final Justification:**

Although I think the authors should perform a more comprehensive comparison aginst [12], from my side, there are no major concerns preventing the acceptance of this paper.

**Limitations:**

yes.

**Paper Formatting Concerns:**

None.

**Quality:**

2

**Strengths And Weaknesses:**

## Strengths

1. **Timely and Impactful Topic**.
   The paper addresses a critical and timely problem—the carbon footprint of large AI models—at a moment when environmental sustainability in AI is garnering significant attention across academia, industry, and policy-making. By focusing on both *operational* and *embodied* carbon emissions, the work reflects a holistic view of AI's environmental impact, aligning with recent calls for more transparent and accountable AI development practices.

2. **Quantitative Evaluation with Clear Metrics**.
   The empirical evaluation is well-structured and includes a comparative benchmark against a widely cited prior method, **LLMCarbon**. The results, indicating a ≤ 7.44% estimation error for PCAM versus ≤ 108.51% for LLMCarbon, are compelling and underscore the effectiveness of the proposed probabilistic modeling approach.

3. **Well-Structured and Coherent Presentation**.
   The paper is clearly written and logically organized. Each section transitions smoothly into the next, and technical components are introduced with sufficient motivation and clarity. This makes the paper accessible not only to sustainability researchers but also to a broader machine learning audience concerned with AI's environmental footprint.


## Weaknesses

1. **No comparison against a highly-related prior work**.
  The paper does not compare PCAM with the method proposed in Zhang et al. [12] ("Spatial-temporal embodied carbon models for the embodied carbon accounting of computer systems", e-Energy 2024). This prior work also addresses uncertainties in embodied carbon and should serve as a critical baseline. I strongly encourage the authors to include a quantitative comparison with [12] in the experimental section. Doing so would more rigorously establish the novelty and practical advantage of PCAM in capturing embodied carbon uncertainty.

---

> ### Author Rebuttal · Authors · 2025-07-31
>
> We thank the reviewer for the insightful feedback. We appreciate your recognition of our paper's contribution to addressing environmental sustainability in large-scale AI. The only concern is the inclusion of an additional baseline. We have added the new baseline in the experiments as requested by the reviewer.
>
> **Weakness[1] and Questions[1]: No comparison against a highly-related prior work. The paper does not compare PCAM with the method proposed in Zhang et al. [12] ("Spatial-temporal embodied carbon models for the embodied carbon accounting of computer systems", e-Energy 2024). ...Could you clarify why this work was not included in your experimental evaluation? Additionally, is there a technical limitation or incompatibility that prevents a direct quantitative comparison between PCAM and [12]? Including such a comparison would significantly strengthen your claims regarding PCAM’s accuracy and novelty.**
>
>
> Thank you for suggesting the inclusion of the STEC model [12] as an additional baseline. We have conducted new experiments comparing PCAM against STEC and add the new experimental results to Table 5 as follows. Overall, the conclusion does change. PCAM still demonstrates a significant advantage in accuracy.
>
> STEC only incorporates spatiotemporal electricity carbon intensity in manufacturing but remains fundamentally deterministic. It fails to probabilistically model the Geotemporal manufacturing variability (regional production capacity + temporal carbon intensity fluctuations) and dynamic manufacturing evolution (yield/fabrication efficiency improvements). To enable the comparison of STEC with PCAM, we extend STEC by scaling its hardware-level accounting to AI models carbon accounting (using Eq.2) and generating the distributions of embodied carbon based on regional electricity carbon intensity frequency data.
>
> The following table compares STEC with PCAM. Specifically, PCAM outperforms STEC consistently across all percentiles, with ≤2.7% error vs. STEC’s ≤23.5% error. Component-wise analysis shows that PCAM outperforms STEC at each component. For CPU carbon accounting, PCAM errors are below 2.6% versus STEC's ≤22.8%. GPU calculations show PCAM’s robust performance with less than 2% deviations across all percentiles, contrasted by STEC’s substantial overestimations reaching 32.3% at the 5th percentile. The DRAM comparison follows similar trends, with PCAM errors remaining below 1.8% in DRAM versus STEC ≤22.2%.
> We omit the comparison of SSD as the extended STEC by us and PCAM use the same approach to model storage. We will add these comparisons to the evaluation section and discuss STEC’s limitations in our revised manuscripts.
>
>
>
> |    Models    | Total Embodied Carbon (kg) at each Percentile |        |         |        |        |
> |:------------:|:---------------------------------------------:|:------:|:-------:|:------:|:------:|
> |              |                      5th                      |  10th  |  Median |  90th  |  95th  |
> | Ground Truth |                     229.2                     | 236.3  |  279.8  | 331.3  | 352.2  |
> | PCAM         |                     235.2                     | 242.7  |  281.1  | 340.0  | 359.9  |
> | PCAM#        |                      2.6%                     |  2.7%  |   0.5%  |  2.5%  |  2.2%  |
> | STEC         |                     283.0                     | 286.0  |  294.9  | 303.0  | 305.0  |
> | STEC#        |                     23.5%                     |  21.0% |   5.4%  |  8.5%  |  13.4% |
> | LLMCarbon    |                     297.7                     |        |         |        |        |
> | LLMCarbon#   |                     23.0%                     |  20.6% |   6.0%  |  11.3% |  18.3% |
> |              |  CPU Embodied Carbon (kg) at each Percentile  |        |         |        |        |
> | Ground Truth |                      1.1                      |  1.2   |   1.5   |  2.0   |  2.2   |
> | PCAM         |                      1.2                      |  1.2   |   1.5   |  2.0   |  2.1   |
> | PCAM#        |                      1.7%                     |  0.8%  |   0.5%  |  1.4%  |  2.6%  |
> | STEC         |                      1.4                      |  1.4   |   1.4   |  1.6   |  1.7   |
> | STEC#        |                     18.4%                     |  16.1% |   4.1%  |  22.0% |  22.8% |
> | LLMCarbon    |                      1.6                      |        |         |        |        |
> | LLMCarbon#   |                     26.5%                     |  23.9% |   4.1%  |  31.5% |  40.3% |
> |              |  GPU Embodied Carbon (kg) at each Percentile  |        |         |        |        |
> | Ground Truth |                     98.7                      | 101.4  |  133.8  | 189.1  | 211.1  |
> | PCAM         |                     97.0                      |  99.4  |  132.3  | 187.6  | 209.4  |
> | PCAM#        |                      1.8%                     |  2.0%  |   1.1%  |  0.8%  |  0.8%  |
> | STEC         |                     130.5                     | 132.7  |  138.8  | 144.8  | 145.9  |
> | STEC#        |                     32.3%                     |  30.9% |   3.8%  |  23.5% |  30.9% |
> | LLMCarbon    |                     141.7                     |        |         |        |        |
> | LLMCarbon#   |                     66.9%                     |  28.5% |   5.6%  |  33.5% |  49.0% |
> |              |  DRAM Embodied Carbon (kg) at each Percentile |        |         |        |        |
> | Ground Truth |                      8.0                      |  8.2   |  10.4   |  14.5  |  15.3  |
> | PCAM         |                      7.8                      |  8.1   |  10.3   |  14.3  |  15.2  |
> | PCAM#        |                      1.8%                     |  1.0%  |   1.3%  |  0.9%  |  0.6%  |
> | STEC         |                      9.7                      |  10.0  |  10.6   |  12.2  |  12.4  |
> | STEC#        |                     22.1%                     |  22.2% |   2.1%  |  15.6% |  18.8% |
> | LLMCarbon    |                     10.9                      |        |         |        |        |
> | LLMCarbon#   |                     26.5%                     |  24.9% |   4.2%  |  33.4% |  41.0% |
> |              |  SSD Embodied Carbon (kg) at each Percentile  |        |         |        |        |
> | Ground Truth |                     106.0                     | 110.2  |  123.6  | 145.2  | 147.7  |
> | PCAM/STEC    |                     108.3                     | 111.5  |  122.8  | 147.1  | 149.7  |
> | PCAM/STEC#   |                      2.1%                     |  1.1%  |   0.7%  |  1.3%  |  1.3%  |
> | LLMCarbon    |                     125.6                     |        |         |        |        |
> | LLMCarbon#   |                     15.6%                     |  12.3% |   1.6%  |  15.5% |  17.6% |
>
> Table: The evaluation results on embodied carbon accounting.

---

### Official Review · Reviewer_Xggk · 2025-06-28

**Clarity:** 3
**Significance:** 3
**Originality:** 3
**Rating:** 5
**Confidence:** 4

**Summary:**

The authors proposed probabilistic carbon accounting model (PCAM) to accurately estimate the carbon footprint of large AI models by capturing the uncertainties in carbon modeling. The uncertainties are: 1) geotemporal manufacturing variability, 2) dynamic manufacturing evolution, 3) dynamic operating context. PCAM will report a distribution instead of a single number that helps make more risk-aware decisions in sustainable AI. They proposed novel approaches to construct distributions for parameters like embodied carbon of processors, embodied carbon of memory, embodied carbon of storage, etc.

To evaluate their proposed method, the authors crafted a dataset including ESG reports, power grid data, etc. They evaluated PCAM on 4 large AI models, XLM, T5, GPT3, Switch, and the results showed that PCAM outperformed LLM Carbon, achieving less than 8% error in both embodied and operational carbon predictions.

**Questions:**

1) Why didn't you just use electricity maps data for operational carbon accounting instead of sampling from a distribution. my question is related to comment 1 on weakness.
2) Can the cooling infrastructure's embodied and operational carbon accounting be modeled with PCAM?
3) How do you think PCAM can be used in practice by AI engineers? How can it be informative and insightful to be able to change the manufacturing process of the parts like processors, memory, etc?

**Ethical Concerns:**

["NO or VERY MINOR ethics concerns only"]

**Final Justification:**

My questions were answered properly by the authors and I keep my ratings the same.

**Limitations:**

The limitation were addressed adequately; however, they can also argue in their discussion about the hidden carbon emissions that is often neglected like labor.

**Quality:**

3

**Strengths And Weaknesses:**

Strengths:
1) Proposed a novel approach for accounting embodied carbon and report a distribution instead of just a number which captures the uncertainty in embodied carbon uncertainty.
2) Significant improvement over the current benchmark (LLM Carbon) in embodied and operational carbon accounting
3) Provide important insight on GPUs are major contributors to carbon uncertainty.
4) The paper is well organized and clear.

Weakness:
1) For operational carbon accounting, the sampling may not capture the temporal characteristics. for example, from 12pm to 2pm in California when solar is abundant, we have low carbon intensity for the electricity, however, by sampling from a distribution, you may get high values of carbon emissions for this time period which makes the operational carbon accounting inaccurate.
2) Other factors like cooling infrastructure in data centers are neglected which plays a pivotal role in carbon accounting

---

> ### Author Rebuttal · Authors · 2025-07-31
>
> We sincerely thank the reviewer for the insightful comments and recognition of the novelty and impact of our work. Below are our responses to the weaknesses and questions.
>
> **Weakness (1) and Question (1): For operational carbon accounting, the sampling may not capture the temporal characteristics. for example, from 12pm to 2pm in California when solar is abundant, we have low carbon intensity for the electricity, however, by sampling from a distribution, you may get high values of carbon emissions for this time period which makes the operational carbon accounting inaccurate. Why didn't you just use electricity maps data for operational carbon accounting instead of sampling from a distribution. my question is related to comment 1 on weakness.**
>
> We apologize for not making our description clear enough. In the evaluation of the operational part, we would like to show the uncertainty/distribution of the carbon footprint of an AI model when the AI model can be trained anywhere/any time. This distribution can provide AI deployers with macroscopic information about the carbon footprint of the model operation. Of course, if the training location or time is specified, this uncertainty will be reduced. As you say, if users want to know the embodied carbon if the training is limited from 12 pm to 2 pm in California, the corresponding carbon intensity data (not using the CI distribution) should be used for the accounting. In fact, our model supports this operation as the carbon intensity is just the input for our model. In our publicly available model code, users can easily change the carbon intensity input according to their needs.
>
> **Weakness (2) and Question(2): Other factors like cooling infrastructure in data centers are neglected which plays a pivotal role in carbon accounting. Can the cooling infrastructure's embodied and operational carbon accounting be modeled with PCAM?**
>
>
> We thank the reviewer for highlighting cooling infrastructure. Its operational carbon is indeed captured via PUE in our model (Eq.7, Table 6). PUE is the ratio of the total amount of energy consumed by the entire data center facility to the energy delivered to the IT equipment.
>
> We analyze the carbon footprint of AI mdoels across four life stages:
> Hardware manufacturing: Emissions from producing core computer components;
> Hardware transport: Emissions from shipping hardware to users;
> Operational use: Emissions from running software (electricity consumption) ;
> End-of-life processing: Emissions from hardware recycling;
> The third stage is related to the operational carbon of AI models. Other stages are related to the embodied carbon of AI models.
>
> Hardware manufacturing and operational use dominate emissions (e.g., >97% for Dell R740), while transport and end-of-life contribute minimally (<3%). We omit transport and end-of-life. Within hardware manufacturing, we focus on computer hardware emissions (e.g., materials, energy, chemicals), omitting ancillary components like buildings, **cooling**, labor. We acknowledge that these are also important for embodied carbon. Quantifying the uncertainties of these requires interdisciplinary collaboration and will be our future work. The main difficulty lies in collecting relevant data. If the related data is available, PCAM's probabilistic framework can be extended to cooling infrastructure using the same distributional modeling approach (e.g., develop the distribution of embodied carbon according to the carbon intensity of electricity used in cooling infrastructure manufacturing). In the absence of data, using a normal distribution is a potential option for quantifying the uncertainty.
>
>
> **Question(3): How do you think PCAM can be used in practice by AI engineers? How can it be informative and insightful to be able to change the manufacturing process of the parts like processors, memory, etc?**
>
>
> PCAM can generate practical value in the following aspects:
>
> (1) Risk-Aware Decision Making & Budgeting:
> PCAM can help an AI company estimate the probability of exceeding specific carbon thresholds or budgets, crucial for realistic sustainability targets and mitigating the risk of policy violations.
>
> (2) Sensitivity Analysis: Identifying which parameters (e.g., operational location/time, hardware components) contribute most to uncertainty. This guides prioritization (e.g., focus on operational scheduling vs. advocating for low-carbon hardware). PCAM can also help hardware manufacturers optimize green production, e.g., factory site selection, capacity scheduling, and carbon-centric process optimization.
>
> (3) Practical Operational Choices: A Cloud computing developer can choose when and where to run models based on the carbon intensity of the electricity grid, since PCAM models these differences over time and place.
>
> (4) Multi-source Hardware Choice: Developers can compare the distribution of embodied carbon for different hardware (e.g., SDD by Samsung vs. SK Hynix) to guide procurement and deployment, balancing performance and environmental impact.
>
> (5) Broader Value & Feasibility:
> While supply-chain factors (like yield) require industry engagement, quantifying their impact motivates advocacy for transparent reporting and sustainable practices;
>
> Probabilistic reporting (confidence intervals) enhances credibility for compliance and stakeholder communication;
>
> PCAM aligns with growing industry trends (e.g., Google's Carbon-Intelligent Computing) by providing the essential confidence bounds needed for robust auditing, reporting, and long-term planning.

---

> > ### Comment · Reviewer_Xggk · 2025-08-02
> >
> > In answering the first question, you mentioned ``In our publicly available model code, users can easily change the carbon intensity input according to their needs.``, would you please elaborate more on how abstract your system is for the user? I mean how much user can change the system architecture and how complex is it for the user, how much domain knowledge does the user need to be able to work with the system?

---

> ### Author Response · Authors · 2025-08-03
>
> Dear Reviewer Xggk,
>
> We sincerely thank you for your feedback. We propose the carbon accounting model (Eq. 2-7) and abstract carbon accounting variables (i.e, EPS, MPS, CI, Y, DieSize, t, T, BD, EPG). For users, the abstracted variables are fixed, while the specific values of the variables are alterable. The entire computation is automatic by the implemented system and thus is easy-to-use. Users only need to know about their basic hardware specifications during the deployment of AI and record the running time. For example, to account for the embodied carbon of an AI model, PCAM requires users to input hardware specifications (Table 4) and the duration for which the AI model operates on the specified hardware. This is the minimum information required to account for the embodied carbon footprint. For other parameters (e.g., CI, Y, EPS, etc.), the user can use our pre-populated database. If the user has more detailed information about the hardware, they may substitute the default values accordingly. For example, the pre-populated distribution of carbon intensity for memory is weighted according to each region’s respective contributions to global DRAM production capacity (Table 2). If the user knows the manufacturing location of the hardware, they can replace the default values with the corresponding carbon intensity data by providing location-specific carbon intensity data via our API, such as CSV files derived from our open-source dataset (Table 3). In this case, the uncertainty of the carbon accounting results will be reduced.

---

> > ### Comment · Reviewer_Xggk · 2025-08-03
> >
> > Thank you for your clarifying answer. I believe this is a well-abstracted theme that the user can understand easily.

---

> > > ### Author Response · Authors · 2025-08-05
> > >
> > > Dear Reviewer Xggk,
> > >
> > > Thank you! We hope you enjoyed reading our work as much as we enjoyed conducting it. And we wish you good fortune in your own papers this year.

---

### Official Review · Reviewer_ug1P · 2025-07-03

**Clarity:** 2
**Significance:** 2
**Originality:** 3
**Rating:** 5
**Confidence:** 4

**Summary:**

The paper reports the development of a new model to estimate the carbon emissions of LLMs. By using a novel dataset assembled by the authors containing semiconductor production data and electricity emissions, the authors estimate parameter distributions for relevant sources of variation in emissions and integrate them into a single predictive model for LLM carbon emissions. The authors report a test of this model compared to LLMCarbon, finding favorable performance.

**Questions:**

•	[Line 63] Why are you reporting results with three or more significant figures, when a substantial part of this work is highlighting uncertainty ranges? This seems to be false precision, and I would recommend no more than 2 sig figs be used.
•	[Line 97] Where did you source the carbon emission factors for Table 1? They are approximately right, but there are significant variations by region and by sub-technology.
•	[Line 141] It’s good that you’ve included the F-gas emissions and the raw material extraction emissions. However, they don’t seem to be addressed in the following discussion. How did you handle those uncertainties?
•	[Line 155] Grids are evolving across the world, and the share of low-carbon electricity is typically increasing in them. Yet you are using data from 2021 and 2023 to estimate carbon intensity of electricity consumption. How can your approach be efficiently kept up-to-date with changing grid generation mixes?
•	[Line 203] The energy efficiency appears to be treated as a variable that improved by a fixed percentage every year, ad infimum. Has this been true of observed efficiency gains in fabs?
•	[Line 247] Your evaluation appears to take published information on training hardware and duration for select LLMs, and convert that into an embodied carbon value. It’s not clear how you’re doing this, or how you’re obtaining a distribution. How do you determine the geographic region where training occurred? How do you determine the emissions associated with manufacturing for hardware used in training? Further, given the input data, this test only speaks to the emissions for training a model, not for inference.
•	[Line 267] Why do you conclude that the large deviation between LLMCarbon and the ground truth data demonstrates a failure to capture uncertainty? The primary conclusion seems to be that LLMCarbon is based on inaccurate or insufficient data. Incorporating uncertainty would presumably not change the median value estimated by LLMCarbon unless distributions of relevant parameters were significantly skewed. Do you think that’s the case?
•	[Line 295] Given geopolitical and economic realities, why do you feel it is feasible for companies training AI models to select different manufacturing timelines and geographic locations for their GPUs? This would appear to be highly impractical.

**Ethical Concerns:**

["NO or VERY MINOR ethics concerns only"]

**Final Justification:**

The authors replied carefully and thoroughly to my comments. They have responded to the three specific issues I raised as needing attention in order to accept for publication. The discussion with other reviewers has also helped clarify that the scope of this paper is a good fit for the venue, which has resolved my concerns in that area. As I result I am changing my rating to accept.

**Limitations:**

Yes

**Paper Formatting Concerns:**

•	[Line 212] “dividing”

**Quality:**

2

**Strengths And Weaknesses:**

Strengths
- Improving the understanding of GHG emissions of LLMs is an important endeavor. The paper suggests ways to better model several different parts of the overall emissions of LLMs, focusing on including uncertainty quantification.
- The authors intend to release a dataset they have assembled with information relevant to performing LCAs for LLMs. From the brief description in the paper, this dataset would seem to be valuable for other LCA practitioners.
- The authors have attempted to test their model against values derived from reported data regarding LLM training.
Weaknesses
- No machine learning is used in the paper. It might be more appropriate for a conference focusing on LCA, supply chain emissions, or data center sustainability.
- Although presented as an LCA analysis, the paper does not include several important aspects of LCA that are specified in ISO 14040. One example is a failure to clearly specify the system boundary. The electricity consumption and IT hardware are included, but the data center physical building and other potential sources of emissions are omitted, without discussion. Another example is a failure to include the full life-cycle of IT components, which encompasses end-of-life disposal.
- The reported evaluation with “ground truth” from published LLM training information is unclear and difficult to follow, but it appears to only compare emissions estimates of the training phase of AI models, not inference.
- The paper strongly asserts the value of uncertainty modeling in LCA. While it’s true that this can provide confidence intervals (as mentioned), that doesn’t translate to enabling AI model developers to make choices about lower-emitting options. For example, yield variation impacts this uncertainty, but there is no practical way AI model developers can directly make business choices to change yield.
- In order to accept this paper for publication, I recommend the following changes. First, a clearer specification of the system boundary and the “life stages” of the LLM that are included / excluded and a discussion of the reasons. Second, a clearer description of how published training data were used to develop a distribution of embodied carbon emission estimates (as “ground truth”), and a clear discussion about whether and how this applies to the inference phase vs the training phase. Third, a clearer discussion about what insights this type of uncertainty-based LCA modeling of LLMs could actually provide to decision-makers developing LLMs, and which of these are practically feasible. With these changes I would be willing to consider revising my recommendation to “accept”.

---

> ### Author Rebuttal · Authors · 2025-07-31
>
> We sincerely appreciate your valuable time and constructive comments.
>
> **W1 and Recommend changes (R)1: Specification of the system boundary and the “life stages” are included/excluded and a discussion of the reasons.**
>
> We analyze the carbon footprint across four life stages:
> Hardware manufacturing: emissions from producing core computer components;
> Hardware transport: emissions from shipping hardware to users;
> Operational use: emissions from running software (electricity consumption) ;
> End-of-life processing: emissions from hardware recycling;
> The third stage is related to the operation carbon of AI models. Other stages are related to the embodied carbon of AI models.
>
> Hardware manufacturing and operational use dominate emissions (e.g., >97% for Dell R740), while transport and end-of-life contribute minimally. We omit transport and end-of-life. Within hardware manufacturing, we focus on computer hardware emissions (e.g., materials, energy, chemicals), omitting ancillary components like buildings, cooling, labor. We acknowledge that these are also important for embodied carbon. Quantifying the uncertainties of these requires interdisciplinary collaboration and will be our future work.
>
> **W2, R2 and Q[L247]: How published training data were used to develop a distribution of embodied carbon emission estimates (as “ground truth”), and a clear discussion about how this applies to the inference phase. How do you determine the geographic region where training occurred? How do you determine the emissions associated with manufacturing for hardware used in training?**
>
> We develop the distribution for hardware and then derive the distribution for AI models based on training duration. To develop the distribution of hardware, we need hardware specs (Tab 4) and hardware manufacturing data (Tab 3). NVIDIA dominates GPU production; thus, we use Taiwan's carbon intensity for GPUs. For multi-sourced components (CPU, SSD, DRAM), the fabrication region is modeled as a discrete random variable. Probabilities are proportional to regional production capacity (e.g., DRAM: US 3%, CN 18%, TW 20%, KR 52%, JP 7%). A mixture CI distribution for each component is constructed by aggregating region-specific CI distributions (via KDE). Similarly, Yield (LINE 174) and EPS (LINE 189) distributions are obtained. Embodied carbon distributions for GPU/CPU (Eq.3), memory(Eq.4), and storage (Eq.5) are then calculated. Other parameters in Eq.3-5 are constants and from the collected hardware dataset (Tab. 3).
>
> The geographic region of training affects operational but not embodied carbon accounting. To reveal the uncertainty, we fix time and vary training regions (90 global locations) for spatial analysis, and fix location (e.g., DE) while varying training times (e.g., throughout 2023) for temporal analysis.
>
> PCAM's probabilistic framework can inherently apply to both training and inference, as it accounts for both based on hardware utilization time and hardware set specifications.
>
> We conducted new experiments on embodied carbon during inference. Specifically, we account for the embodied carbon of ChatGPT-3 inference based on the published inference data from [1]. The published inference data contains average computation per request (i.g, 2.07 GPU-seconds /request) and the corresponding hardware set (i.g, NDasrA100_v4 sizes series).
>
> Due to the word limit of the rebuttal, we list a simplified version of the experimental results table below (the complete one will be in the revised manuscript). Overall, our conclusion does not change. PCAM still outperforms LLMCarbon, demonstrating great alignment with ground truth values across all percentiles, maintaining a maximum deviation of only 2.7% at the 10th percentile. In contrast, LLMCarbon shows substantially larger deviations up to 41%.
>
> |Error|Toatal|CPU|GPU|DRAM|SSD|
> |-|-|-|-|-|-|
> |PCAM|<2.7%|<2.3%|<2.2%|<1.8%|<2.6%|
> |LLMCarbon|<33%|<31%|<41%|<36%|<24%|
>
> **W3,R3: discussion about what insights could actually provide to decision-makers developing LLMs, and which of these are practically feasible**
>
> PCAM can generate practical value in the following aspects.
>
> (1)Risk-Aware Decision Making & Budgeting:
> PCAM can help an AI company estimate the probability of exceeding specific carbon thresholds or budgets, which is crucial for realistic sustainability targets and mitigating the risk of policy violations (e.g., carbon tax, carbon credit).
>
> (2)Sensitivity Analysis: Identifying which parameters (e.g., operational location/time, hardware components) contribute most to uncertainty. This guides prioritization (e.g., focus on operational scheduling vs. advocating for low-carbon hardware).
>
> (3)Practical Operational Choices: A Cloud computing developer can choose when and where to run models to reduce operational carbon based on the carbon intensity of grids, since PCAM models these differences over time and place.
>
> (4)Multi-source Hardware Choice: Developers can compare the distribution of embodied carbon for different hardware (e.g., SDD by Samsung vs. SK Hynix) to guide green risk-aware procurement and deployment, balancing performance and environmental impact.
>
> (5)Broader Value & Feasibility:
> While supply-chain factors (like yield) require industry engagement, quantifying their impact motivates advocacy for transparent reporting and sustainable practices.
>
> Probabilistic reporting (confidence intervals) enhances credibility for compliance and stakeholder communication.
>
> PCAM aligns with growing industry trends (e.g., Google's Carbon-Intelligent Computing) by providing the essential confidence bounds needed for robust auditing, reporting, and long-term planning.
>
> **Q[L63] Comments on significant figures**
>
> We sincerely appreciate your comments. We will revise that in our revised manuscript.
>
> **Q[L97] Where did you source the carbon emission factors for Table 1? They are approximately right, but there are significant variations by region and by sub-technology**
>
> They are from ref[2]. The factors are used as model inputs. Our published code allows users to easily replace them with more precise regional or technology-specific factors if available.
>
> **Q[L141] Uncertainties about the F-gas and the raw material**
>
> We acknowledge these uncertainties were not quantified in the discussion. We use the average values from industrial reports [30][31]. Quantifying their uncertainty relies on data disclosed by manufacturers. We do not find relevant data in manufacturers' ESG reports. In the absence of historical data, using a normal distribution is a potential option for quantifying uncertainty.
>
> **Q[L155] ...You are using data from 2021 and 2023 to estimate the carbon intensity of electricity consumption. How can your approach be efficiently kept up-to-date with changing grid generation mixes?**
>
> Since hardware typically has a 5-year lifespan, much of it in use is not newly manufactured. Additionally, our model takes carbon intensity as an input, and our public code allows users to easily input the latest grid carbon intensity data or specify values based on the model’s training or hardware manufacturing time.
>
> **Q[L203] The energy efficiency appears to be treated as a variable that improved by a fixed percentage every year, ad infimum. Has this been true of observed efficiency gains in fabs?**
>
> The energy efficiency data is from TSMC’s ESG reports [20]. It improves every year, but as the mass production time increases, the improvement rate will become smaller. For example, the normalized energy efficiency of 16nm improves annually as (1.3, 1.7, 2.8, 3.2, 3.3).
>
> **Q[L267] Why do you conclude that the large deviation between LLMCarbon and the ground truth data demonstrates a failure to capture uncertainty? The primary conclusion seems to be that LLMCarbon is based on inaccurate or insufficient data. Incorporating uncertainty would presumably not change the median value estimated by LLMCarbon unless distributions of relevant parameters were significantly skewed. Do you think that’s the case?**
>
> LLMCarbon intrinsically considers a hardware class (e.g., 28nm CPU) to have the same embodied carbon. Yet a hardware instance in this class can be manufactured from diverse regions and in diverse time periods with different embodied carbon, e.g., in the 2021 winter in Ireland or in the 2022 summer in Taiwan with different CI, Y, and EPS. LLMCarbon uses the average data of the hardware class. PCAM models the distribution of the hardware class. The average of PCAM output should be close to the output of LLMCarbon.
>
> **Q[L295] The feasibility of companies choosing different manufacturing schedules and geographies for their GPUs.**
>
> We agree that GPU production is geographically constrained due to NVIDIA’s technical monopoly. However, memory/storage components have lower technical barriers and multiple suppliers (e.g., Samsung, SK Hynix, Seagate). Their embodied carbon rivals GPUs. While currently challenging commercially, evolving carbon policies (carbon taxes, trading systems) will increasingly equate emissions with cost when carbon carries direct financial weight; some options that were not feasible before will become feasible.
>
> **W4: The scope of this paper**
>
> Our work addresses the urgent issue of large AI models’ carbon footprint at a time when environmental sustainability in AI is a major concern across academia, industry, and policy. By examining uncertainties in both operational and embodied carbon emissions, our work offers a comprehensive perspective on AI’s environmental impact, supporting recent demands for greater transparency and accountability in AI development.
>
> **Formatting Concern[L212]:**
>
> We have carefully reviewed and revised the formatting throughout the manuscript.
>
> Ref:
>
> [1]Chien, Andrew A., et al. "Reducing the Carbon Impact of Generative AI Inference (today and in 2035)." HOTCARBON. 2023.
>
> [2]Maji, Diptyaroop., et al. "CarbonCast: multi-day forecasting of grid carbon intensity." ACM Buildsys. 2022.

---

> > ### Comment · Reviewer_ug1P · 2025-08-08
> >
> > Thank you for the thoughtful and thorough reply to my comments.
> >
> > Regarding the system boundary and life-cycle stages, I appreciate the clarification. Your approach does seem reasonable as described, for the scope of the current work. As you note, future interdisciplinary work could include more attention to other emissions sources (buildings, cooling, etc) that have been excluded from the system boundary here, although I don't anticipate those contributing significantly. (Please note also that an LCA practitioner might like to see a more formalized presentation of the "functional unit" that you're analyzing.)
> >
> > Regarding the development of ground truth data and choice of geographic regions, thank you for the detailed comments and additional experiments. This has clarified most of my question. I do think there is a challenge in using the Taiwan grid carbon intensity to understand GPU production because of the complications of renewable energy procurement - a fab that sources low-carbon electricity via PPA would apparently not be given any emissions "benefits" for that under your approach. However, LCA best practice is not always clear on how renewable energy sourcing should be included, so I don't have a specific suggestion for improvement. I recommend this as an interesting area of future research.
> >
> > Regarding the utility of your model to provide insights to decision-makers developing LLM, I agree with the points about practical operational choices and sensitivity analysis. I am more skeptical about the potential for hardware supply changes, given the time lag between procurement and inference (as you note) and the exclusion of electricity procurement choices, among other factors.
> >
> > My other questions have also been satisfactorily answered. Given the discussion with other reviewers, I am comfortable with the subject of this paper being a fit for this venue. I will be changing my rating to accept.

---

> > > ### Author Response · Authors · 2025-08-08
> > >
> > > Dear Reviewer ug1P,
> > >
> > > Thank you very much for taking the time to review our response and for your positive feedback. We will incorporate your valuable suggestions into the revised manuscript. We are committed to retaining these improvements, as they significantly enhance the quality of our paper. Thank you again for your time and effort, and wish you all the best in your research and professional endeavors.
> > >
> > > Sincerely,
> > >
> > > Authors #28102

---

> ### Author Response · Authors · 2025-08-05
>
> Dear Reviewer ug1P,
> ﻿
>
> We sincerely thank you for the time and effort you have dedicated to reviewing our paper. We deeply appreciate your contributions, especially given the demands on your time.
> ﻿
> We are truly enthusiastic about this work and its findings, and we look forward to the upcoming discussion. Please do not hesitate to ask any questions or request further clarification. We are happy to provide them.
> ﻿
>
> Sincerely,
> Authors #28102

---

### Official Review · Reviewer_ETm3 · 2025-07-15

**Clarity:** 2
**Significance:** 3
**Originality:** 2
**Rating:** 5
**Confidence:** 2

**Summary:**

This paper presents PCAM, a Probabilistic Carbon Accounting Model for estimating the carbon footprint of large AI models. It addresses a key limitation in prior work like LLMCarbon by modeling uncertainty in both embodied and operational carbon—specifically due to geo-temporal variability, manufacturing evolution, and operating context. PCAM uses KDE-based parameter distributions and a carbon dataset compiled from ESG reports and energy statistics. Compared to LLMCarbon, PCAM achieves significant error reductions. The work is well-motivated and timely in light of increasing interest in AI sustainability.

**Questions:**

In terms of quantification method and evaluation, some important points are not clear to me:
Questions:
- It is not clear how’s the ground truth embodied carbon data is calculated in Table 5?
- Section 4.3 “We estimate the electricity consumption of AI models based on the training data” -> How’s this estimation done? i.e. How’s the “Ave. Power” of the different models calculated in Table 6?
- What’s the geographic granularity used for operational carbon analysis in the US? US is much larger compared to Germany/Italy geographic and there is a lot of differences between each Balancing Authorities in the US. In Figure 5, is the variability only caused by temporal differences? If so it means data is averaged significantly for US and that’s probably the reason for why the density function has less variance compared to other countries. Operational carbon estimation tools such as ElectricityMaps provide finer granularity than country level geographic granularity.

**Ethical Concerns:**

["NO or VERY MINOR ethics concerns only"]

**Final Justification:**

The authors addressed majority of my questions/concerns. This is a high‑quality paper tackling an important and timely problem in carbon accounting, using AI as a primary use-case. Given the prior precedence of similar papers appearing in top ML venue, I would increase my score to accept.

**Limitations:**

Yes

**Quality:**

3

**Strengths And Weaknesses:**

**Strengths**
- Important problem: The paper tackles a meaningful and underexplored issue—quantifying uncertainty in AI carbon footprint estimation—which has practical implications for sustainability-aware system design and policy compliance.
- Novel contribution: Introduces a probabilistic modeling approach (PCAM) that goes beyond deterministic estimates, enabling risk-aware decision-making.
- Dataset contribution: Authors commit to release a comprehensive dataset for embodied carbon estimations along with the code

**Weaknesses**
- Lack of component attribution analysis: While the paper models distributions over multiple factors (e.g., yield, carbon intensity), it does not perform an attribution analysis to assess which parameter contributes most to overall variance. Such sensitivity analysis would be valuable to guide optimization or intervention.

- Ground truth ambiguity: In Table 5, it's unclear how the "ground truth" embodied carbon values were calculated. Are these empirical measurements, model-derived, or based on re-aggregated estimates?

- Power estimation unclear: Section 4.3 mentions estimating electricity consumption from training data, but the method for computing “Ave. Power” in Table 6 is not described. Is it measured, averaged across hardware, or estimated from spec sheets?

- Geographic granularity concerns: The operational carbon analysis seems to aggregate carbon intensity at the country level. For a geographically diverse region like the US, this averaging may obscure intra-country variance. Figure 5 suggests lower variance for the US, but it’s unclear if this is due to temporal-only variation or over-averaging. Tools like ElectricityMaps offer balancing-authority-level granularity that may yield more accurate results.

- Hardware lifespan effects: While lifespan is modeled in the formulas, its role in uncertainty or sensitivity is not discussed in the evaluation. Additionally, environmental factors (e.g., temperature, humidity) that affect hardware degradation and embodied carbon are not modeled.

- Typos and presentation issues: There are a number of typos and formatting issues (e.g., “Trainin Day” in Table 6), which detract from polish and clarity.

- Fit for NeurIPS: While the work is high-quality, it may be better suited for venues in systems, sustainability, or hardware design (e.g., HPCA, ISCA, or HotCarbon), as the paper’s core technical contribution lies in accounting methodology rather than ML algorithmics.

---

> ### Author Rebuttal · Authors · 2025-07-31
>
> We sincerely appreciate the reviewer's insightful comments. We are glad that the reviewer found our work solved an important problem and made a Novel contribution both in approach and dataset. Below, we respond to each of your points individually.
>
> **W(1):Lack of component attribution analysis: While the paper models distributions over multiple factors, it does not perform an attribution analysis to assess which parameter contributes most to overall variance...**
>
> We have now conducted a Sobol sensitivity analysis for embodied carbon of XLM model for each hardware component. Below, we summarize key results.
>
> (1)Hardware Contribution Analysis:
> We first analyze the relative contribution of hardware components to total embodied carbon as the following table shows. We find that GPUs (50%) and SSDs (46%) dominate the embodied carbon footprint of the AI model, collectively accounting for 96% of emissions. CPUs (1%) and DRAM (4%) play minor roles comparatively. This confirms GPUs and SSDs as primary optimization targets.
>
> | Hardware | CPU | GPU | SSD | DRAM |
> |-|-|-|-|-|
> | EC (%)   |  1% | 50% | 46% |   4% |
>
>
> (2)Parameter Sensitivity Attribution:
> The blow table shows the component attribution analysis for each hardware using the Sobol approach. We find that:
>
> GPU: All parameters significantly impact variance, with Yield (Y, 0.97) and Carbon Intensity (CI, 0.99) being the strongest drivers;
>
> CPU: CI (0.99) and EPS (0.98) dominate over Yield (0.85). This is mainly because, compared to a GPU, a CPU uses a more mature process (a more stable yield).
>
> SSD: Carbon Intensity (CI, 1.0) is the exclusive determinant of variance;
> DRAM: Both CI (0.98) and EPS (0.97) contribute near-equally to uncertainty;
>
> | Sobol Indices | Y     | CI   | EPS  |
> |-|-|-|-|
> | CPU           | 0.85  | 0.99 | 0.98 |
> | GPU           |  0.97 | 0.99 | 0.95 |
> | SSD           | /     |    1 | /    |
> | DRAM          | /     | 0.98 | 0.97 |
>
> Key Insight from Analysis:
>
> (1)Shift Focus to Storage Optimization:
>
> Storage contributes 46% of embodied carbon—nearly equal to GPUs (50%)—yet remains overlooked in sustainability efforts. Beyond GPU-centric strategies, AI developers can prioritize:
> (a)Reducing storage footprint (e.g., via data-efficient training, model pruning).
> (b)Adopting low-capacity/high-durability SSDs to minimize replacement cycles.
>
> (2)Procurement Options:
>
> While NVIDIA’s market dominance limits sustainable options for GPUs (e.g., fab location), for storage, a diverse range of suppliers (Seagate, Samsung, Western Digital) supports green sourcing to reduce the embodied carbon of AI deployments (e.g., prioritizing SSDs from regions and time periods with lower CI).
>
>
> **W(2) and Q(1): Ground truth ambiguity: In Table 5, it's unclear how the "ground truth" embodied carbon values were calculated. Are these empirical measurements, model-derived, or based on re-aggregated estimates?
> It is not clear how’s the ground truth embodied carbon data is calculated in Table 5?**
>
> We apologize for the unclear description. The “ground truth” is based on the published LLM training information (e.g., training duration, hardware set), the hardware manufacturing data set we collected (see Table 3). We further clarify as follows.
>
> The published LLM training information used for the embodied carbon accounting in Table 5 contains (1) training duration of the AI model $t$ (i.g., 20.4 days for XLM in Table 5); (2) the hardware set used to train the model, e.g., GPU(12nm), CPU (16nm), Storage (64*32TB), Memory(64*256gb), as shown in Table 4.
>
> We clarify that the carbon intensity of the manufacturing process used in the calculation of embodied carbon uses the following data:
> (1)GPU: NVIDIA has a monopoly, so production is located in Taiwan, but the manufacturing period is uncertain. Therefore, we use recent carbon intensity data (2021-2023) from Taiwan.
> (2)CPU (16nm), Storage (64*32TB), and Memory (64*256GB) have multiple manufacturers to choose from. We aggregate the carbon intensity of the manufacturing process based on the wafer fabrication capacity of each global region at the corresponding technology node (Table 2).
>
> We can get the distribution of Yield and EPS for each hardware according to its technology node, e.g., 16nm CPU, 12nm GPU, 10nm DDR4 DRAM (see Figures 3 and 4), from the hardware manufacturing data set we collected (see Table 3).
>
> Now, we have all the distributions in Eqs. 3, 4, and 5. Then we can get the embodied carbon of the AI model using Eq.2. We label this result as “ ground truth” in the evaluation. In the evaluation, the output of PCAM is the accounting result using resampled data based on the probability distribution curve generated by PCAM via KDE.
>
>
> **W(3) and Q(2): Power estimation unclear: Section 4.3 mentions estimating electricity consumption from training data, but the method for computing “Ave. Power” in Table 6 is not described. Is it measured, averaged across hardware, or estimated from spec sheets?
> Section 4.3 “We estimate the electricity consumption of AI models based on the training data” -> How’s this estimation done? i.e. How’s the “Ave. Power” of the different models calculated in Table 6?**
>
> We apologize for this unclear description. The electricity consumption is calculated based on published training information from [8][1].“avg. system power (W)” conveys the average system power per computing device, including TPU/GPU, host CPU, DRAM, and network interface. It is measured data. The electricity consumption is calculated by Training time * Ave. Power *Num. of device *PUE.
>
>
> **W(4) and Q(3): Geographic granularity concerns: ...For a geographically diverse region like the US, this averaging may obscure intra-country variance. Figure 5 suggests lower variance for the US, but it’s unclear if this is due to temporal-only variation or over-averaging. ...What’s the geographic granularity used for operational carbon analysis in the US? ...In Figure 5, is the variability only caused by temporal differences?**
>
> We sincerely thank the reviewer for raising this important point about geographic granularity in US operational carbon analysis. You are correct that aggregating carbon intensity at the country level obscures significant intra-country variance in large, diverse regions like the US, and that Balancing Authority (BA)-level data (e.g., from ElectricityMaps) offers finer resolution. In our initial analysis (Fig 5), operational carbon for the US was indeed calculated using national average grid carbon intensity. We acknowledge this likely suppressed observable variance. We have performed a new experiment using hourly carbon intensity data for California (CAISO) from ElectricityMaps, representing BA-level granularity. Results comparing CAISO to the US national average are summarized below:
>
> | Operational   Carbon (t) | 5th per. | 10th per. | Median | 90th per. | 95th per. |  Var.  |
> |:-:|:-:|:-:|:-:|:-:|:-:|:-:|
> |          US          |   33.4   |   34.7    |  39.8  |   45.1    |   46.4    | 15.8  |
> |         CAISO        |   12.9   |   14.6    |  26.1  |   33.8    |   35.4    | 49.8  |
>
>
>
>
> Key Observations:
>
> The significantly higher variance (49.8 vs 15.8) and wider spread (e.g., 5th percentile: 12.9 vs 33.4) in CAISO confirm that country-level averaging masks critical regional variability in the US.
>
>
> **W(5): Hardware lifespan effects: While lifespan is modeled in the formulas, its role in uncertainty or sensitivity is not discussed in the evaluation. Additionally, environmental factors (e.g., temperature, humidity) that affect hardware degradation and embodied carbon are not modeled.**
>
> PCAM is the first framework to quantify uncertainty in AI’s carbon footprint, focusing on the uncertainty caused by spatiotemporal characteristics in manufacturing. While PCAM incorporates hardware lifespan (T) in its probabilistic model, T is currently treated as deterministic (fixed at 5 years). We plan to add a sensitivity analysis to assess how variations in T affect embodied carbon uncertainty. Preliminary results indicate that reducing T to 4 years increases amortized carbon by up to 25%, whereas extending T to 6 years decreases it by approximately 17%.
>
> We clarify the system boundary of PCAM encompasses four LCA stages: hardware manufacturing, hardware transport, operational use (software electricity consumption), and end-of-life processing (hardware recycling). Operational use corresponds to operational carbon, while the other stages pertain to embodied carbon of AI models.
>
> Hardware manufacturing and operational use dominate emissions (e.g., >97% for Dell R740), while transport and end-of-life contribute minimally (<3%). We omit transport and end-of-life. Within hardware manufacturing, we focus on computer hardware emissions in manufacturing process (e.g., materials, energy, chemicals), omitting ancillary components like buildings, cooling, labor, **environmental factors **. We acknowledge that these are also important for embodied carbon. Quantifying the uncertainties of these requires interdisciplinary collaboration and will be our future work. The main difficulty lies in collecting relevant data. In the absence of data, using a normal distribution is a potential option for quantifying the uncertainty.​
>
> **W(6):Typos and presentation issues...**
>
> We sincerely thank the reviewer for their meticulous reading of our manuscript. We have undertaken a thorough, line-by-line proofreading of the entire manuscript.
>
> **W(7): The scope of this paper.**
>
> Our work addresses the urgent issue of large AI models’ carbon footprint at a time when environmental sustainability in AI is a major concern across academia, industry, and policy. By examining uncertainties in both operational and embodied carbon emissions, our work offers a comprehensive perspective on AI’s environmental impact, supporting recent demands for greater transparency and accountability in AI development.

---

> > ### Comment · Reviewer_ETm3 · 2025-08-03
> >
> > Thank you for the detailed response and for sharing new results. Many of my concerns have been addressed. This is a high‑quality paper tackling an important and timely problem in carbon accounting, using AI as a primary use-case.
> >
> > I remain hesitant about the venue fit, so I will maintain my borderline‑accept recommendation.

---

> > > ### Author Response · Authors · 2025-08-03
> > >
> > > Dear Reviewer ETm3,
> > >
> > > Thank you for your positive feedback. We appreciate your recognition of the paper’s quality and the importance of addressing timely issues in AI carbon accounting, as well as your acknowledgment that our responses have resolved many of your initial concerns.
> > >
> > > **Regarding the remaining hesitation about venue fit:** We understand this concern and would like to briefly emphasize why we believe the core contributions of this work are a strong fit for the NeurIPS community:
> > >
> > > We respectfully highlight that our baseline work, LLMCarbon (the recent acceptance of the highly relevant work [1], ICLR 2024 Oral), underscores the ML community's recognition of the importance of accounting methodology contributions in the carbon footprint of large AI models, even when they are not centered on AI algorithmics. LLMCarbon successfully demonstrated the value of tackling challenges in accounting methodology of carbon footprint for sustainable AI at a premier ML venue. Building on this, our work further addresses the uncertainty in large AI model carbon footprint accounting, provides an open-source accounting model, and develops a comprehensive carbon dataset by aggregating from multiple sources (industrial reports, ESG reports, LCA reports, etc.), offering a complementary and significant advancement to sustainable AI.
> > >
> > > We appreciate your time and thoughtful consideration of our response. We respect your expertise and final judgment regarding the paper's suitability for NeurIPS, regardless of the outcome.

---

### Decision · Program_Chairs · 2025-09-17

**Decision:**

Accept (poster)

**Comment:**

This paper presents a Probabilistic Carbon Accounting Model (PCAM) providing a carbon footprint distribution for large AI models. In PCAM, the footprint is broken down in several contributions and uncertainties pertaining to 1) geotemporal manufacturing variability, 2) dynamic manufacturing evolution, and 3) dynamic operating context are propagated. The model relies on a curated hardware and electricity dataset constructed by aggregating data from diverse sources (including Environmental, Social, and Governance (ESG) reports from hardware manufacturers, power grid operator statistics, industry reports, and peer-reviewed research publications, as detailed in Table 3), that the authors promised to open-source when the paper is published. From this data set, Kernel Density Estimators are derived, and random variable can be propagated though the equations underlying PCAM.   The paper was refereed by four reviewers with gradings ranging from weak accept to accept (one weak accept, three accept gradings). The paper was found to be well written and of practical relevance, yet several questions and concerns were raised in the reviews and discussion. Overall, the referees acknowledged post discussion to be satisfied with the answers and/or that their concerns were addressed.I too found the paper to be of interest and prone to generate valuable discussions in a conference such as NeurIPS, yet I was puzzled (in line with ETm3) by the employed notion of ground truth. While I conceive that it might not be feasible to get some real carbon footprint measurements for validation purposes, it is not entirely clear to me to what extent the model used to generate the said ground truth parallels the PCAM model and more precisely whether one is comparing here a marginal distribution with a conditional one (e.g., with NVIDIA being produced in Taiwan). I find this to call for further clarifications in the paper regarding what the ground truth distributions precisely stand for (whether the ground truth terminology is kept or not). On a different note, I found the idea of using PCAM in a sensitivity analysis context very interesting. This might require some care though about potential independence assumptions between the “parameters”, in case classical versions of variance-based global sensitivity analysis are used, and special procedures dealing with dependent factors, else. I hope this is useful towards finalizing this paper.